# First Synthesis of Racemic *Trans* Propargylamino-Donepezil, a Pleiotrope Agent Able to Both Inhibit AChE and MAO-B, with Potential Interest against Alzheimer’s Disease

**DOI:** 10.3390/molecules26010080

**Published:** 2020-12-27

**Authors:** Benjamin Guieu, Cedric Lecoutey, Rémi Legay, Audrey Davis, Jana Sopkova de Oliveira Santos, Cosimo Damiano Altomare, Marco Catto, Christophe Rochais, Patrick Dallemagne

**Affiliations:** 1Normandie Univ, UNICAEN, CERMN, 14000 Caen, France; benjamin.guieu@unicaen.fr (B.G.); cedric.lecoutey@unicaen.fr (C.L.); remi.legay@unicaen.fr (R.L.); audrey.davis@unicaen.fr (A.D.); jana.sopkova@unicaen.fr (J.S.d.O.S.); 2Department of Pharmacy-Drug Sciences, University of Bari Aldo Moro, 70125 Bari, Italy; cosimodamiano.altomare@uniba.it (C.D.A.); marco.catto@uniba.it (M.C.)

**Keywords:** Alzheimer’s disease, acetylcholinesterase, monoamine oxidase, donepezil, rasagiline, MTDL

## Abstract

Alzheimer’s disease (AD) is a multifactorial neurodegenerative disease towards which pleiotropic approach using Multi-Target Directed Ligands is nowadays recognized as probably convenient. Among the numerous targets which are today validated against AD, acetylcholinesterase (ACh) and Monoamine Oxidase-B (MAO-B) appear as particularly convincing, especially if displayed by a sole agent such as ladostigil, currently in clinical trial in AD. Considering these results, we wanted to take benefit of the structural analogy lying in donepezil (DPZ) and rasagiline, two indane derivatives marketed as AChE and MAO-B inhibitors, respectively, and to propose the synthesis and the preliminary in vitro biological characterization of a structural compromise between these two compounds, we called propargylaminodonepezil (PADPZ). The synthesis of racemic *trans* PADPZ was achieved and its biological evaluation established its inhibitory activities towards both (*h*)AChE (IC_50_ = 0.4 µM) and (*h*)MAO-B (IC_50_ = 6.4 µM).

## 1. Introduction

Among the therapeutic agents getting to clinical trials, more and more examples illustrate the concept of Multi-Target Directed Ligands (MTDLs) [1]. Such compounds display several activities by interacting with different biological targets that are implied in a given disease, in order to obtain a synergy of action. MTDLs appear to be particularly interesting against multifactorial diseases such as Alzheimer’s Disease (AD). As a matter of fact, the classical pharmacological approach consisting in interacting very selectively with a single target has shown clinical limitations, failing to restore such complex biological systems. Thus, the association of drugs has been favored for many years now. Comparatively to this approach, the advantages of MTDLs, in addition to the synergistic action they theoretically display, are the absence of drug–drug interactions, and a better patient compliance by reducing the number of intakes [2].

Basically, there are two main types of MTDLs: conjugates, containing two different active moieties linked by a chemical bond, and molecules that are able to interact indiscriminately with several therapeutic targets, combining the structural elements needed for both activities. As an example, our group recently described donecopride, first MTDL inhibiting acetylcholinesterase (AChE) and activating serotoninergic receptors 5-HT_4_, and now in preclinical evaluation [3,4,5].

Ladostigil (Figure 1) is a novel type of MTDL currently in phase II clinical trials. It is the first compound able to release an active compound, the hydroxyrasagiline, a Monoamine Oxidase B (MAO-B) inhibitor, through the interaction with a first target, the AChE, which is temporarily inhibited during the process [6].

Besides the originality of the mechanism of action of ladostigil, the first results of its evaluation seem to validate the interest lying in targeting both AChE and MAO-B.

AChE inhibition is the mechanism of action of the main anti-AD drugs, leading to a symptomatic effect by preventing acetylcholine (ACh) degradation, responsible for AD symptoms. The efficiency of these drugs, however, decreases with the evolution of the pathology. Associating the latter with a neuroprotective effect appears theoretically able to express a disease-modifying effect while maintaining a long-acting symptomatic benefit.

MAO-B is already an identified target for Parkinson’s disease. The enzyme recycles dopamine in the brain, leading to the formation of hydrogen peroxide, particularly neurotoxic. Its inhibition then provides a neuroprotective effect, which is currently evaluated [7]. More recently, MAO-B was also identified as implied in the amyloid cascade, comforting the therapeutic interest of this target for AD [8,9,10].

Taking into consideration the interest lying in the association into a sole structure of activities directed towards both AChE and MAO-B in AD [8], we aimed at conceiving a new MTDL targeting these enzymes but contrarily to ladostigil in an indifferently manner, i.e., able to bind to each of the active site of these targets. To achieve such a goal, the latter have to share sufficient common features liable to be recognized by a same agent. In support of such allegation, we consider the fact that both donepezil (DPZ), a marketed AChE inhibitor (AChEI), and rasagiline, a marketed, MAO-B inhibitor (MAO-BI), possess an indane core, and we designed propargylaminodonepezil (PADPZ) as a perfect structural compromise between these two drugs. This paper describes our efforts to achieve the first synthesis of PADPZ and its preliminary biological evaluation towards these two enzymes.

## 2. Results

### 2.1. Chemistry

The synthetic strategy we followed for the access to PADPZ (Scheme 1), involved an aldolisation reaction between a properly *N*-protected dimethoxyindanone and *para*-*N* benzylpiperidinecarboxal-dehyde. The methylene derivatives would have then to successively undergo, in various order, a reduction of its alkene moiety, a deprotection of its amino group and a substitution of the latter by a propargyl group.

The access to aminodimethoxyindanones (**3**, **5**–**7**) involved, in a first step, a Rodionov-Johnson synthesis of aminodimethoxyphenylpropionic acid (**2**), starting from veratraldehyde (**1**), followed by a one-pot protection/cyclization reaction, using trifluoroacetic acid and anhydride, to give the trifluoracetylaminoindanone (**3**) in 65% yield, according to a previous reported sequence (Scheme 2) [9,10]. Hydrolyzing **3** by aqueous hydrochloric acid yielded the hydrochloric acid salt (**4**) in 76% yield. The latter was then converted into the *N*-propargyl (**5**), -acetyl (**6**) and *tert*butyloxycarbonyl (**7**) derivatives with 38%, 76% and 83% yields, respectively.

At the same time, *para N*-benzylpiperidine carboxaldehyde (**11**) was obtained in 45% overall yield, through the *N*-benzylation of ethyl 4-piperidinecarboxylate (**8**), followed by a reduction of the ester (**9**) into the alcohol (**10**) and a final Swern oxidation of the latter (Scheme 3) [11,12,13].

In a first attempt, aldolisation of **3** was undertaken in acidic medium but failed to give the methylene derivative (**12**) (Scheme 4). The latter was obtained in alkaline medium, quasi exclusively under its *E* form. The latter was attributed through a selective 1D NOE experiment during which a response in the signal of the proton, located in the para position of the nitrogen atom of the piperidine ring, was observed upon excitation of H3, accounting for a close position between the two protons. Further, a significant deshielding of the signal of the methylene proton was also observed for the *E* form (6.70 ppm) of **12**, due to the cone-shaped shielding zone of the carbonyl group, versus those of the *Z* form (6.25 ppm) observed as a trace.

All attempts, however, to hydrolyze the trifluoroacetylamino group of **12** in alkaline medium failed. Such a chemical behavior was already observed with aminoindanones which, in these conditions, are suitable for an internalization of their double bond likely to lead to a degradation [14]. This can also explain the low yield observed in the preparation of **12** (19%).

All attempts aiming at synthesizing the propargyl derivative (**13**) starting from **5** and according to a similar manner, failed, probably for the same reasons.

In a similar manner, the *N*-acetyl and *N*-Boc methylene derivatives (**14**) and (**15**) were obtained starting from **6** and **7**, exclusively under their *E* form and in 51% and 46% yield respectively (Scheme 5). This geometry was attributed by analogy with the structure of **12**. Reduction of the latter succeeded this time, using NaBH_4_ for **14** and H_2_ Pd/C for **15**, respectively. Only one diastereoisomeric form of the *N*-acetylcompound **16** was recovered, explaining the moderate yield observed (35%). Its *trans* geometry was established by its X-ray diffractometry study (Figure 2). The two diastereoisomeric forms of the *N*-boc derivative **17** were obtained, with a relative proportion of 3 *trans* for 1 *cis* and a global yield of 65%. Attribution of the two forms of **17** was achieved through 1D NOE experiment which showed, upon excitation of the NH proton, a response observed on the signal of H2 for the *trans* form and on the signal of the CH_2_ protons for the *cis* form, respectively. It was not possible to separate them. The coupling constant values between H2 and H3 signals appeared higher in the *cis* form (6.5 Hz) versus the *trans* one (3.5 Hz) and confirmed the major formation of *trans*-**17** by comparison with the ^1^H-NMR spectrum of *trans*-**16**.

All the attempts aiming at deprotecting the *N*-acetyl group of **16** failed. Compound **18**, however, was obtained in 78% yield starting from the *N*-Boc derivative (**17**), which was easily *N*-deprotected using TFA. Compound **18** was recovered as a mixture of its *trans*/*cis* forms in a 90/10 ratio, from which the *trans* form was isolated.

Finally, the expected PADPZ (**19**) was successfully obtained under its *trans* form, through the *N*-substitution of *trans*-**18** by a propargyl group.

### 2.2. In Silico Results

#### 2.2.1. Molecular Modeling Study

The aim of the present molecular modeling work was to check whether PADPZ (**19**) can bind to the active sites of the two targeted proteins and which of the two enantiomers of its *trans* form will be the more prone to display the more effective interaction with both AChE and MAO-B. The initial 3D model of compound **19** was built using the structurally closed X-ray structure of compound **16**, its acetylamino analog. Two enantiomers were present in the crystal of **19**, and therefore, the two enantiomers of *trans*-**19** (*R*,*R* and *S*,*S*) were built. The models, further, were protonated on their piperidine nitrogen.

Firstly, the two enantiomers of **19** were docked separately into the (*h*)AChE active site and in a comparative manner with donepezil (Figure 3A). In a similar manner as in our previous studies on (*h*)AChE [4], a water molecule interacting with protonated piperidine ring of donepezil was conserved (residue number 931) during docking. The results suggested different binding affinities for the two enantiomers. The enantiomer *R*,*R* (Figure 3B) took an orientation within the active site of AChE closed to donepezil one, accounting for a possible inhibitory activity towards the enzyme similar to those of the reference compound. This enantiomer, indeed, reproduced crucial donepezil interactions: (i) the charged nitrogen of the piperidine ring was oriented in a position suitable for an interaction with the water molecule in the proximity of Tyr337 and Tyr341, (ii) the carbonyl group formed a hydrogen bond with NH of the Phe295 backbone and (iii) both benzene rings are placed in parallel to the Trp279 and Trp84 indole rings to favor a π-stacking interaction. An additional interaction is further observed between the piperidine NH of **19** and the OH of Tyr124.

On the other hand, the position of the enantiomer *S*,*S* of **19** within the AChE groove did not appear as favorable for an inhibitory interaction (Figure 3C). It lost some crucial interactions through the protonated piperidine nitrogen of **19**, the π-stacking with Trp84 and the hydrogen bond between the carbonyl group of **19** and the NH of Phe295. Contrariwise this carbonyl group established a hydrogen bond with Tyr124.

To check the ability of the enantiomers of *trans*-**19** to bind to the (*h*)MAO-B binding site, a covalent docking was carried into the enzyme. The docked enantiomers were previously modified, engaging their propargylamino group into a covalent bond with the FAD co-factor of MAO-B and modifying the bond order in a same manner as observed in rasagiline-FAD complex (Figure 4A). The covalent docking was then carried out and both enantiomers were placed into the MAO-B cavity without any steric clash (Figure 4B,C). The interactions established by the two enantiomers with MAO-B are very similar: a hydrogen bond between their carbonyl group and the Gln206 side chain and another one between their methoxy groups and a tyrosine residue, Tyr326 for *R*,*R* and Tyr188 for *S*,*S*. From these observations, we hypothesized that the two enantiomers should bind to MAO-B with similar affinities and inhibitory activities.

#### 2.2.2. In Silico ADME Parameters

In silico ADME parameters for PADPZ have been calculated using SwissADME (www.SwissADME.ch). No alert was identified (Table 1).

### 2.3. In Vitro Results

#### Cholinesterases and Monoamine Oxidases Inhibition

The inhibitory activity of *trans*-PADPZ (**19**) as well as its acetylamino analogue (**16**) towards (*h*)AChE and (*eq*)BuChE was evaluated according to the Ellman test [15]. DPZ was used as a reference. Results are depicted in Table 2.

The inhibitory activity of *trans*-PADPZ (**19**) towards human recombinant MAO-A and -B was determined as already described [17], measuring the fluorescence of 4-hydroxyquinoline produced by MAOs in the oxidative deamination of substrate kynuramine (50 µM). Rasagiline and pargiline were used as references, as well as the dimethoxyketo analogue of rasagiline (**5**) we synthesized. Results are depicted in Table 3.

## 3. Discussion

The modulation of DPZ and indanone moiety to develop novel AChEI has been largely explored in recent years [19]. Indeed, the binding site of the enzyme could accommodate to an important number of chemical structures, and several MTDL have been obtained starting from an indanone moiety. Indeed, several modulations have generated MTDL with Aβ-aggregation [20] or BACE1 [21] inhibition. During these SAR, the presence of α-β insaturated ketone has generally led to a decrease in potency compared to DPZ. If most of the modulation has been engaged by changing the nature of the piperidine or replacing the dimethoxy substituent, few examples have been described in introducing substituent on the indanone moiety. With this aim, we have prepared and focus our attention on the preparation of the reduced form of PADPZ. *Trans*-PADPZ (**19**) showed a noticeable inhibitory activity towards (*h*)AChE with an IC_50_ = 440 nM and a relative selectivity towards (*h*)BuChE which was inhibited at a concentration 10-fold greater (IC_50_ = 4.2 µM). This AChE inhibitory activity, however, appeared weaker than those of DPZ (IC_50_ = 14 nM), but higher than those of the acetamide *trans*-**16** (% inhibition at 10^−6^ M = 4%) which almost totally lost its effect.

Tested against (*h*)MAO-B, *trans*-PADPZ exhibited again a sound activity with an IC_50_ = 6.4 µM, in a same order of magnitude than those of pargiline (IC_50_ = 2.7 µM), but weaker with respect to rasagiline (IC_50_ = 14 nM) used as references. Interestingly, *trans*-PADPZ (**19**) appeared selective towards MAO-B with a weak inhibition of MAO-A (% inhibition at 10^−6^ M = 13%). The dimethoxyketo analogue of rasagiline (**5**) displayed also a modest and selective inhibitory activity towards MAO-B (IC_50_ = 13.4 µM). According to the docking study, this inhibition is not influenced by the stereochemistry of the substitution at the difference of the corresponding rasagiline or ladostigil.

*Trans*-PADPZ (**19**) appears therefore as a dual AChE/MAO-B with a good selectivity towards BuChE and MAO-A. These results are consistent with the predictive data issued from the in silico study, which further gave us an incentive to separate the enantiomers of *Trans*-PADPZ, since one of them (*R*,*R*) appeared, according to this study, more able to inhibit AChE than the other one (*S*,*S*).

## 4. Materials and Methods

### 4.1. Chemistry

#### 4.1.1. General Methods

All chemical reagents and solvents were purchased from commercial sources and used without further purification. Melting points were determined on a STUART SMP50 melting point apparatus (Cole-Parmer, Vernon Hills, IL, USA) ^1^H, ^13^C and ^19^F-NMR spectra were recorded on a BRUKER AVANCE III 400 MHz (Bruker, Billerica, MA, USA) with chemical shifts expressed in parts per million downfield from TMS as an internal standard and coupling in Hertz. IR spectra were recorded on a Perkin-Elmer BX FT-IR apparatus (Perkin Elmer, Wellesley, MA, USA) using KBr pellets. High resolution mass spectra (HRMS) were obtained by electrospray on a BrukermaXis (Bruker, Billerica, MA, USA). The purities of all tested compounds were analyzed by LC−MS, with the purity all being higher than 95%. Analyses were performed using a Waters Alliance 2695 (Waters corporation, Milford, MA, USA) as separating module (column XBridge C18 2.5 M/4.6 X 50 mM (Waters corporation, Milford, MA, USA) using the following gradients: A (95%)/B (5%) to A (5%)/B (95%) in 4.00 min. This ratio was hold during 1.50 min before return to initial conditions in 0.50 min. Initial conditions were then maintained for 2.00 min (A = H_2_O, B = CH_3_CN; each containing HCOOH: 0.1%). MS were obtained on a SQ detector by positive ESI.

#### 4.1.2. Synthesis of Compounds (**2**, **3**, **5**–**11**)

*3-Amino-3-(3,4-dimethoxyphenyl)propanoic acid* (**2**) [12,13]. To a solution of 3,4-dimethoxy benzaldehyde (1.0 eq., 6 g, 36.1 mmol) in EtOH (200 mL), were added malonic acid (2.0 eq., 7.52 g, 72.2 mmol) and ammonium acetate (2.0 eq., 5.57 g, 72.2 mmol). The mixture was stirred to reflux overnight. After cooling to room temperature, the reaction mixture was filtered in vacuo and washed by hot EtOH to give **2** as a white solid (6.11g, 75%). M.p: 235 °C; ^1^H NMR (400 MHz, D_2_O) *δ* 7.16–6.85 (m, 3H), 4.57 (dd, *J* = 8.2, 6.5 Hz, 1H), 3.84 (s, 3H), 3.82 (s, 3H), 2.87 (dd, *J* = 16.2, 8.2 Hz, 1H), 2.76 (dd, *J* = 16.2, 6.5 Hz, 1H); ^13^C-NMR (101 MHz, D_2_O) *δ* 177.2, 148.5, 148.2, 128.9, 120.0, 111.9, 110.4, 55.6, 55.6, 52.5, 40.3; IR (neat, cm^−1^): 3411 (N-H), 2933, 2636 (C-H), 2124 (C=C), 1571, 1549 (C=O), 1398, 1148; LC-MS *m*/*z* [M − H]^+^ 224.35.

*N-(5,6-Dimethoxy-3-oxo-indan-1-yl)-2,2,2-trifluoroacetamide* (**3**) [12,13]. 3-Amino-3-(3,4-dimethoxy phenyl)propanoic acid (7.60 g, 33.74 mmol) was stirred in trifluoroacetic acid (3 mL·mmol^−1^), and trifluoroacetic anhydride was added (3 mL·mmol^−1^). The mixture was refluxed for 2 h. After cooling at room temperature, the reaction was stopped by adding slowly 10 mL of water. The mixture was evaporated in vacuo. The crude was dissolved in saturated aqueous solution of NaHCO_3_ (400 mL) and EtOAc (400 mL); then, the organic layer was dried over MgSO_4_ and concentrated in vacuo. The compound **3** was obtained as a yellow powder (6.61 g, 65%). M.p: 219 °C; ^1^H NMR (400 MHz, CD_3_OD) *δ* 7.19 (s, 1H), 7.08 (s, 1H), 5.59 (dd, *J* = 7.5, 3.0 Hz, 1H), 3.93 (s, 3H), 3.88 (s, 3H), 3.15 (dd, *J* = 18.8, 7.5 Hz, 1H), 2.57 (dd, *J* = 18.8, 3.0 Hz, 1H); ^13^C NMR (100 MHz, CD_3_OD) *δ* 202.2, 156.5, 151.2, 146.6, 146.4, 129.9, 121.1, 106.5, 103.3, 55.4, 55.1, 47.9, 44.2; ^19^F-NMR (376 MHz, CD_3_OD) *δ*-77.23; IR (neat, cm^−1^) *ν* max 3303 (N-H), 3095, 2835, 1694 (C=O), 1594, 1269, 1048, 857; LC-MS *m*/*z* [M + H]^+^ 304.35.

*5,6-Dimethoxy-1-oxoindan-3-ylammonium chloride* (**4**) [12,13]. A solution of **3** (6 g, 0.02 mol) in aqueous HCl 6N (50 mL) was refluxed for 30 min, cooled and filtered. The filtrate was evaporated to dryness under reduced pressure. The solid residue was recrystallized form Et_2_O to give **4** (3.8 g, 78%) as colorless crystal. M.p. 208 °C; ^1^H-NMR (400 MHz, DMSO-d_6_) *δ* 9.00 (br s, 3H), 7.91 (s, 1H), 7.13 (s, 1H), 4.83 (m, 1H), 3.90 (s, 3H), 3.84 (s, 3H), 3.04 (dd, *J* = 18.5, 6.8 Hz, 1H), 2.60 (dd, *J* = 18.5, 3.4 Hz, 1H); IR (neat, cm^−1^) *ν* max 3100–2700 (^+^NH_3_), 1680 (C=O), 1325.

*5,6-Dimethoxy-3-(prop-2-ynylamino)indan-1-one* (**5**). Propargyl bromide (326 µL, 4.3 mmol) and K_2_CO_3_ (560 mg, 4 mmol) were added to a solution of **4** (500 mg, 2 mmol) in acetonitrile. The reaction mixture was stirred at 80 °C for 5 h. The solvent was removed under reduced pressure, and the residue was purified by column chromatography on silica using CH_2_Cl_2_/MeOH (100/2) as eluent to give **5** as a brown solid (190 mg, 38%). M.p. 112 °C; ^1^H-NMR (400 MHz, CDCl_3_) *δ* 7.15 (s, 1H), 7.08 (s, 1H), 4.53 (dd, *J* = 6.5, 2.7 Hz, 1H), 3.98 (s, 3H), 3.90 (s, 3H), 3.63–3.45 (m, 2H), 2.97 (dd, *J* = 18.5, 6.5 Hz, 1H), 2.55 (dd, *J* = 18.5, 2.8 Hz, 1H), 2.30 (t, *J* = 2.4 Hz, 1H); ^13^C NMR (101 MHz, CDCl_3_) δ 202.8, 155.6, 150.5, 150.4, 129.9, 106.8, 103.8, 81.9, 72.2, 56.4, 56.2, 55.0, 44.7, 36.7; IR (neat, cm^−1^) *ν* max 3426 (N-H), 2923, 1694 (C=O), 1594, 1500, 1311, 1268; LC-MS *m*/*z* [M + H]^+^ 191.02.

*N-(5,6-Dimethoxy-3-oxo-indan-1-yl)acetamide* (**6**). An aqueous solution of **3** (360 mg, 1.18 mmol,) in HCl 6N was refluxed for 24 h. After evaporation, a suspension of the obtained amino salt in THF was cooled to 0 °C, and triethylamine (493 µL, 3.54 mmol) and acetic anhydride (251 µL, 2.66 mmol) were added. The mixture was stirred at room temperature for 4 h, then filtered and concentrated under reduced pressure. Purification on silica gel using CH_2_Cl_2_/EtOAc (80/20) as eluent provided **6** as a white solid (76%). ^1^H NMR (400 MHz, CDCl_3_) *δ* 7.14 (s, 1H), 7.01 (d, *J* = 0.7 Hz, 1H), 5.90–5.74 (m, 1H), 5.64 (dddd, *J* = 8.2, 7.3, 3.0, 0.8 Hz, 1H), 3.98 (s, 3H), 3.91 (s, 3H), 3.20 (dd, *J* = 19.0, 7.4 Hz, 1H), 2.42 (dd, *J* = 19.0, 3.0 Hz, 1H), 2.07 (s, 3H).

Tert*-Butyl N-(5,6-dimethoxy-3-oxo-indan-1-yl)carbamate* (**7**). Boc_2_O (671 mg, 3.075 mmol) and TEA (415 mg, 4.1 mmol) were added to a solution of **4** (500 mg, 2.05 mmol) in CH_2_Cl_2_. The reaction mixture was stirred at room temperature for 24 h and then diluted with water. Extraction with EtOAc, washing of the organic layer with brine and evaporation to dryness gave a residue. Purification on the latter by column chromatography on silica using CH_2_Cl_2_/EtOAC (80/20) as eluent gave **7** (530 mg, 83%) as a colorless solid. ^1^H NMR (400 MHz, CDCl_3_) *δ* 7.15 (s, 1H), 7.05 (s, 1H), 5.44–5.20 (m, 1H), 4.84 (d, *J* = 8.8 Hz, 1H), 3.98 (s, 3H), 3.91 (s, 3H), 3.18 (dd, *J* = 18.9, 7.3 Hz, 1H), 2.45 (d, *J* = 18.9 Hz, 1H), 1.49 (s, 9H); ^13^C-NMR (101 MHz, CDCl_3_) *δ* 202.0, 156.0, 155.8, 150.9, 149.4, 129.8, 106.8, 103.8, 80.3, 56.5, 56.4, 48.7, 45.5, 28.5 (3C);. HRMS/ESI: *m*/*z* calcd. [M+Na]^+^ 330.1317, found 330.1320.

*Ethyl 1- benzylpiperidine-4-carboxylate* (**9**) [14,15,16]. To a solution of ethyl piperidine-4-carboxylate (3 g, 19.1 mmol, 1.0 eq.) in CH_2_Cl_2_ (300 mL,) was added benzyl bromide (2.72 mL, 22.92 mmol, 1.2 eq.) and triethylamine (3.09 mL, 22.92 mmol, 1.2 eq.), at 0 °C. After 1 night at room temperature; the mixture was washed with saturated aqueous solution of NaHCO_3_. The organic layer was washed with brine and dried over MgSO_4_. After filtration and concentration in vacuo, purification on silica gel using cyclohexane/EtOAc (35:65) as an eluant provided an orange oil (3.75g, 79%). ^1^H NMR (400 MHz, CDCl_3_) *δ* 7.37–7.30 (m, 4H), 7.30–7.22 (m, 1H), 4.15 (q, *J* = 7.1 Hz, 2H), 3.51 (s, 2H), 2.88 (dt, *J* = 12.1, 3.9 Hz, 2H), 2.30 (tt, *J* = 11.2, 4.1 Hz, 1H), 2.04 (td, *J* = 11.4, 2.7 Hz, 2H), 1.97–1.86 (m, 2H), 1.86–1.70 (m, 2H), 1.27 (t, *J* = 7.1 Hz, 3H); ^13^C-NMR (101 MHz, CDCl_3_) δ 175.3, 138.4, 129.1 (2C), 128.2 (2C), 127.0, 63.3, 60.3, 53.0 (2C), 41.3, 28.3 (2C), 14.2; LC-MS *m*/*z* [M + H]^+^ 248.60.

*(1-Benzyl-4-piperidyl)methanol* (**10**) [14,15,16]. To a suspension of LiAlH_4_ (2.57g, 67.7 mmol, 4eq.) in 40 mL of THF was added dropwise a solution of **9** (4.2 g, 16.9 mmol, 1.0 eq.) in 40 mL of THF. The mixture was heated to reflux for 3h and was then slowly quenched with 80 mL of ethyl acetate and 25 mL of NaOH 1N solution followed by 30 mL of water to afford a granular inorganic precipitate. After the night, the solution was filtered through celite; the solid was washed with EtOAc, and then, the filtrate was concentrated under reduced pressure to give reduced compound as a primrose yellow oil (3.1g, 89% yield). ^1^H NMR (CDCl_3_, 399.8 MHz) *δ* 7.35–7.28 (m, 4H), 7.28–7.22 (m, 1H), 3.50 (s, 2H), 3.48 (d, *J* = 6.5 Hz, 2H), 2.97–2.86 (m, 2H), 2.00–1.92 (m, 3H, H9), 1.75–1.64 (m, 2H), 1.57–1.42 (m, 1H), 1.35–1.21 (m, 2H); ^13^C-NMR (CDCl_3_, 100.5 MHz) *δ* 138.0, 129.5 (2C), 128.2 (2C), 127.1, 67.6, 63.5, 53.5 (2C), 38.5, 28.7 (2C).

*1-Benzylpiperidine-4-carbaldehyde* (**11**) [14,15,16]. A solution of oxalyl chloride (2.6 mL, 30.2 mmol, 2.0 eq.) in CH_2_Cl_2_ was cooled down to −48 °C, then DMSO (2.36 mL, 33.2 mmol, 2.2 eq.) was added. After 10 min, a solution of the alcohol **9** (3.1 g, 15.1 mmol, 1.0 eq.) in CH_2_Cl_2_ was added dropwise. After 15 min, Et_3_N (8.66 mL, 64.2 mmol, 4.2 eq.) was added and the mixture was brought back to room temperature. Water was then added, and the organic phase was washed with brine, then dried on MgSO_4_, filtered and evaporated under reduced pressure. Purification on alumina gel (CH_2_Cl_2_) provided the expected aldehyde **10** as a brown oil (60%). ^1^H-NMR (400 MHz, CDCl_3_) *δ* 9.60 (d, *J* = 1.3 Hz, 1H), 7.36–7.14 (m, 5H), 3.45 (s, 2H), 2.77 (dt, *J* = 11.8, 4.0 Hz, 2H), 2.19 (ttd, *J* = 10.7, 4.2, 1.3 Hz, 1H), 2.06 (td, *J* = 11.1, 2.9 Hz, 2H), 1.90–1.81 (m, 2H), 1.64 (dtd, *J* = 14.2, 10.7, 3.8 Hz, 2H); ^13^C-NMR (101 MHz, CDCl_3_) δ 204.1, 138.3, 129.1 (2C), 128.2 (2C), 127.1, 63.3, 52.5 (2C), 48.0, 25.5 (2C).

#### 4.1.3. General Procedure for the Aldolization Reaction and Preparation of Compounds (**12**, **15**)

To a solution of the ketone (1 eq.) in EtOH (500 µL/mmol) were added an aqueous solution of NaOH 5N (250 µL/mmol) and a solution of the aldehyde **10** (1.3 eq.) in EtOH. The mixture was stirred at room temperature for 5 h and then concentrated under reduced pressure. Water was added, and the reaction was extracted with EtOAc. The organic layer was washed with brine and dried over MgSO_4_ and then filtered and concentrated under reduced pressure.

*N-[(2E)-2-[(1-Benzyl-4-piperidyl)methylene]-5,6-dimethoxy-3-oxo-indan-1-yl]-2,2,2-trifluoroacetamide* (**12**). The reaction was performed according to the general procedure. The residue was purified by column chromatography on silica using CH_2_Cl_2_/MeOH (90/10) as eluent, then on alumina using CH_2_Cl_2_/EtOAc (80/20) as eluent to provide **12** as an orange solid (19%) which was separated from **15**. M.p. 199 °C; ^1^H-NMR (400 MHz, CDCl_3_) *δ* 7.48–7.27 (m, 4H), 7.14 (s, 1H), 7.00–6.97 (m, 1H), 6.97 (s, 1H), 6.72 (dd, *J* = 10.8, 1.7 Hz, 1H), 6.17 (d, *J* = 9.0 Hz, 1H), 3.98 (s, 3H), 3.91 (s, 3H), 3.61 (s, 2H), 2.04 (t, *J* = 10.4 Hz, 2H), 2.41–2.38 (m, 1H), 2.18–1.55 (m, 6H); ^13^C-NMR (101 MHz, CDCl_3_) *δ* 190.1, 157.2 (q, *J* = 37.7 Hz, CF_3_), 156.4, 151.2, 144.8, 143.4, 137.8, 135.4, 131.2, 129.3 (2C), 128.4 (2C), 127.3, 115.9 (q, *J* = 287.7 Hz, CF_3_), 106.6, 104.3, 63.3, 56.7, 56.1, 52.7, 52.6, 48.5, 37.1, 31.1, 31.0; IR (neat, cm^−1^) *ν* max 3435 (N-H), 2922, 2850, 1716 (C=O), 1689, 1648, 1549, 1211, 1182; LC-MS *m*/*z* [M + H]^+^ 488.81.

Tert*-Butyl N-[(2E)-2-[(1-benzyl-4-piperidyl)methylene]-5,6-dimethoxy-3-oxo-indan-1-yl]carbamate* (**15**). The reaction was performed according to the general procedure. Yellow solid (46%). ^1^H-NMR (400 MHz, CDCl_3_) *δ* 7.36–7.27 (m, 5H), 7.23 (s, 1H), 7.06 (s, 1H), 6.70 (dd, *J* = 10.5, 1.8 Hz, 1H), 5.83 (d, *J* = 9.9 Hz, 1H), 4.72 (d, *J* = 9.8 Hz, 1H, NH), 3.98 (s, 3H), 3.93 (s, 3H), 3.55–3.46 (m, 2H), 3.01–2.81 (m, 2H), 2.61–2.46 (m, 1H), 2.17–1.91 (m, 2H), 1.78–1.52 (m, 4H), 1.50 (s, 9H); ^13^C-NMR (100 MHz, CDCl_3_) *δ* 190.9, 156.1, 155.7, 150.9, 146.1, 143.8, 138.4, 137.2, 131.0, 129.3 (2C), 128.3, 127.1 (2C), 106.9, 104.3, 80.3, 63.6, 56.6, 56.4, 53.0 (2C), 49.7, 36.4, 31.5, 31.4, 28.6 (3C). IR (neat, cm^−1^) *ν* max 3380, 2933, 1701 (C=O), 1500, 1296; LC-MS *m*/*z* [M + H]^+^ 492.83; HRMS/ESI: *m*/*z* calcd. [M + H]^+^ 493.2702, found 493.2703.

#### 4.1.4. Synthesis of Compounds (**14**, **16**–**19**)

*N-[(2E)-2-[(1-benzyl-4-piperidyl)methylene]-5,6-dimethoxy-3-oxo-indan-1-yl]acetamide* (**14**). A solution of **6** (200 mg, 0.8 mmol) in MeOH (1 mL) was heated at 50 °C. At this temperature, K_2_CO_3_ (774 mg, 5.6 mmol) was added dropwise to 500 µL of water over 30 min. After 20 min, a solution of **11** (196 mg, 0.96 mmol) in MeOH was added. After stirring for 2 h 30 min, the mixture was concentrated under vacuo; then, water was added, and an extraction was performed using EtOAc. The organic phase was washed with a brine and dried on MgSO_4_ and then filtered and concentrated under reduced pressure. After a column on neutral alumina using CH_2_Cl_2_/EtOAc (80:20) as eluent, another purification on silica gel using EtOAc/MeOH (90/10) as eluent was performed to provide the expected compound as a white solid (176 mg, 51%). ^1^H-NMR (400 MHz, CDCl_3_) *δ* 7.40–7.30 (m, 4H), 7.29–7.22 (m, 1H), 7.00 (s, 1H), 6.95 (s, 1H), 6.60 (dd, *J* = 10.6, 1.8 Hz, 1H), 6.39 (dd, *J* = 9.4, 3.0 Hz, 1H, NH), 6.14 (dd, *J* = 9.2, 1.7 Hz, 1H), 3.96 (s, 3H), 3.83 (s, 3H), 3.51 (d, *J* = 4.0 Hz, 2H), 2.99–2.90 (m, 2H), 2.46 (dddd, *J* = 17.9, 10.9, 6.9, 4.1 Hz, 1H), 2.13–1.97 (m, 5H), 1.73–1.53 (m, 4H); ^13^C-NMR (101 MHz, CDCl_3_) *δ* 191.1, 169.7, 156.1, 150.6, 145.8, 143.7, 138.3, 137.0, 130.8, 129.2 (2C), 128.2 (2C), 127.0, 106.8, 103.8, 63.5, 56.6, 56.0, 53.0, 52.8, 48.0, 36.7, 31.5, 31.2, 23.4.

*Trans N-[2-[(1-Benzyl-4-piperidyl)methyl]-5,6-dimethoxy-1-oxo-indan-4-yl]acetamide* (**16**). To a solution of **14** (230 mg, 0.53 mmol) in MeOH was added in small portions NaBH_4_ (60 mg, 1.59 mmol). The reaction mixture was concentrated under reduced pressure; then, water was added, and it was extracted with EtOAc. The organic phase was washed with a brine and dried over MgSO_4_ and then filtered and concentrated under reduced pressure. Purification on neutral alumina gel using dichloromethane MeOH (95/5) as eluent provided the expected product as a colorless solid (80 mg, 35%). ^1^H-NMR (400 MHz, CDCl_3_) *δ* 7.34–7.28 (m, 4H), 7.25–7.21 (m, 1H), 7.09 (s, 1H), 6.95 (s, 1H), 5.81 (d, *J* = 9.1 Hz, 1H, NH), 5.36 (dd, *J* = 9.1, 3.2 Hz, 1H), 3.95 (s, 3H), 3.89 (s, 3H), 3.48 (s, 2H), 2.87 (t, *J* = 11.1 Hz, 2H), 2.47 (ddd, *J* = 9.8, 5.1, 3.3 Hz, 1H), 2.07 (s, 3H), 1.96 (m, 2H), 1.83 (m, 1H), 1.73–1.58 (m, 3H), 1.53 (m, 1H), 1.37–1.25 (m, 2H); ^13^C NMR (101 MHz, CDCl_3_) *δ* 204.1, 169.8, 156.1, 150.8, 147.9, 138.6, 129.2 (2C), 128.8, 128.2 (2C), 126.9, 106.7, 103.7, 63.4, 56.5, 56.2, 53.7, 53.7, 53.6, 53.5, 37.4, 33.4, 32.9, 31.8, 23.5; LC-MS *m*/*z* [M + H]^+^ 436.96.

Tert*-Butyl N-[2-[(1-Benzyl-4-piperidyl)methyl]-5,6-dimethoxy-3-oxo-indan-1-yl] carbamate* (**17**). To a stirring solution of **15** (0.41 mmol,1.0 eq.) in EtOH (10 mL) was added 10% palladium on carbon (0.02 g, 50 mg/mmol). Hydrogen gas was then pumped into the reaction vessel at 1 atm and left to stir for 48 h to 40 °C. The catalyst was then removed by filtering the reaction mixture through celite. The solvent was then removed by rotary evaporation to afford the crude product which was purified by column chromatography to yield the reduced product as a colorless solid (65%) with a mixture of *trans*/*cis* (74/26). *Trans*
^1^H-NMR (400 MHz, CDCl_3_) *δ* 7.33–7.28 (m, 4H), 7.26–7.22 (m, 1H), 7.11 (s, 1H), 6.98 (s, 1H), 4.99 (dd, *J* = 9.9, 3.5 Hz, 1H), 4.82 (d, *J* = 9.9 Hz, 1H, NH), 3.96 (s, 3H), 3.90 (s, 3H), 3.51 (s, 2H), 2.88 (m, 3H), 2.02 (m, 2H), 1.86 (m, 1H), 1.74–1.54 (m, 4H), 1.48 (s, 9H), 1.33 (m, 2H); ^13^C NMR (101 MHz, CDCl_3_) *δ* 204.0, 155.9, 155.6, 150.7, 148.0, 138.3, 129.3 (2C), 128.8, 128.1 (2C), 127.0, 106.5, 103.7, 80.1, 63.3, 56.4, 56.2, 55.1, 53.9, 53.6 (2C), 37.2, 33.2, 32.9, 31.6, 28.4 (3C); *Cis*
^1^H-NMR (400 MHz, CDCl_3_) *δ* 7.33–7.28 (m, 4H), 7.26–7.22 (m, 1H), 7.12 (s, 1H), 7.00 (s, 1H), 5.42 (dd, *J* = 10.1, 6.5 Hz, 1H), 4.62 (d, *J* = 10.1 Hz, 1H, NH), 3.97 (s, 3H), 3.90 (s, 3H), 3.51 (s, 2H), 2.89 (m, 2H), 2.48 (m, 1H), 2.02 (m, 2H), 1.86 (m, 1H), 1.74–1.54 (m, 4H), 1.48 (s, 9H), 1.33 (m, 2H); ^13^C-NMR (101 MHz, CDCl_3_) *δ* 204.9, 155.8, 155.6,150.7, 147.9, 138.3, 129.3 (2C), 128.7, 128.1 (2C), 127.0, 106.8, 103.7, 80.1, 63.3, 56.5, 56.2, 53.6 (2C), 51.5, 48.3, 33.5, 32.9, 32.5, 32.1, 28.4 (3C); LC-MS *m*/*z* [M + H]^+^ 494.85.

*Trans 3-Amino-2-[(1-benzyl-4-piperidyl)methyl]-5,6-dimethoxy-indan-1-one* (**18**). To a solution of **17** (500 mg, 1.01 mmol, 1.0 eq.) in CH_2_Cl_2_ was added TFA (2 mL/mmol). The mixture was stirred at room temperature overnight. After completion, the reaction was basified with an aqueous solution of NaOH 1N and then extracted with CH_2_Cl_2_. The organic phase was washed with brine, dried over MgSO_4_, filtered and evaporated under reduced pressure. Reversed-phase flash chromatography (Acetonitrile/water) provided the expected primary amine as an orange oil (orange oil, 78%). ^1^H-NMR (400 MHz, CDCl_3_) *δ* 7.32–7.26 (m, 4H), 7.25–7.20 (m, 1H), 7.11 (s, 1H), 7.04 (s, 1H), 4.06 (d, *J* = 3.7 Hz, 1H), 3.97 (s, 3H), 3.89 (s, 3H), 3.49 (s, 2H), 2.89 (m, 2H), 2.36 (m, 1H), 1.97 (m, 2H), 1.89–1.76 (m, 2H), 1.69–1.59 (m, 2H), 1.46 (m, 1H), 1.33 (m, 2H). ^13^C-NMR (100 MHz, CDCl_3_) *δ* 204.9, 155.8, 151.1, 150.3, 138.5, 129.2 (2C), 128.5, 128.1 (2C), 126.9, 106.1, 103.7, 63.4, 57.3, 57.0, 56.4, 56.2, 53.7 (2C), 37.3, 34.0, 33.0, 32.1.

*Trans 2-[(1-Benzyl-4-piperidyl)methyl]-5,6-dimethoxy-3-(prop-2-ynylamino)indan-1-one* (**19**). Propargyl bromide (35 µL, 0.38 mmol) and K_2_CO_3_ (42 mg, 0.31 mmol) were added to a solution of **16** (110 mg, 0.28 mmol) in acetonitrile. The reaction mixture was stirred at room temperature for 24 h. The solvent was removed under reduced pressure, and the residue was purified by column chromatography on silica using EtOAc/MeOH/NH_4_OH (97/3/41) as eluent to give **14** as a colorless solid (7 mg, 6%). ^1^H-NMR (400 MHz, CDCl_3_) *δ* 7.35–7.27 (m, 4H), 7.26–7.21 (m, 1H), 7.14 (s, 1H), 7.04 (s, 1H), 4.20 (d, *J* = 2.1 Hz, 1H), 3.99 (s, 3H), 3.91 (s, 3H), 3.62–3.43 (m, 4H), 2.96–2.83 (m, 2H), 2.66–2.57 (m, 1H), 2.31 (t, *J* = 2.4 Hz, 1H), 1.98 (t, *J* = 11.5 Hz, 2H), 1.80–1.62 (m, 4H), 1.56–1.48 (m, 1H), 1.47–1.17 (m, 2H);. ^13^C NMR (100 MHz, CDCl_3_) *δ* 205.6, 155.6, 150.6, 148.8, 138.5, 129.2, 129.2 (2C), 128.1 (2C), 126.9, 106.9, 104.0, 82.0, 72.3, 63.4, 61.7, 56.4, 56.2, 53.8, 53.7, 52.3, 38.5, 36.1, 33.7, 32.7, 32.5; HRMS/ESI: *m*/*z* calcd. [M + H]^+^ 433.2491, found 433.2507; LC-MS *m*/*z* [M + H]^+^ 432.87.

### 4.2. X-ray Diffractometry

Single X-ray crystal analysis on compound **16** was carried out on BRUKER D8 Quest diffractometer (Bruker, Billerica, MA, USA) with a PHOTON II detector (Bruker, Billerica, MA, USA) using 1 µS micro focus X-ray source (Mo Kα, λ = 0.71073 Å). The crystal structure was solved by direct methods and refined employing full-matrix least-squares refinement against F2 using SHELX2014 package [22]. All non-hydrogen atoms were refined anisotropically and hydrogen atom positions were determined via difference Fourier maps and refined with isotropic atomic displacement parameters. The structural data of compound **16** have been deposited with Cambridge Crystallographic Data Center, the CCDC (Deposition Number 2052041).

### 4.3. In Silico Study

The 3D models of compound **19** were built using both enantiomers present in the solved X-ray structure of similar compound **16**. Their protonation states at pH 7.4 were predicted using standard tools of the ChemAxon Package (http://www.chemaxon.com/) and the majority microspecies protonated on piperidine nitrogen at this pH were used for docking studies.

For docking into human AChE, its crystallographic coordinates were retrieved from X-ray structure of the donepezil/AChE complex (PDB ID 4EY7, a structure refined to 2.35 Å with an R factor of 17.7%) [23].

The docking of the compound **19** into AChE was carried out with the GOLD program (v5.7.2) using the default parameters [24,25]. This program applies a genetic algorithm to explore conformational spaces and ligand binding modes. To evaluate the proposed ligand positions, the ChemPLP fitness function was used. The binding site in the (*h*)AChE model was defined as a 7 Å sphere from the co-crystallized donepezil ligand and a water molecule interacting with protonated piperidine ring of donepezil was conserved during the docking (residue number 931).

The starting structure for our docking into the (*h*)MAO-B was the high-resolution (2.2 Å) X-ray structure of rasagiline-inhibited human monoamine oxidase B in complex with 2-(2-benzofuranyl)-2-imidazoline (PDB ID 2XFQ) [26]. The rasagiline and 2-(2-benzofuranyl)-2-imidazoline were removed before docking. As the compound **19** should bind covalently to the FAD co-factor through the propargyl moiety in the same way as rasagaline, the structure of **19** was modified by adding the nitrogen atom on the propargyl group and modifying its double bonds repartition according to the one observed in the rasagaline-(*h*)MAO-B complex. The covalent docking constraint between the nitrogen atom of FAD group (N5) and added nitrogen on propargyl moiety was applied during the docking. The binding site in the MaO-B model was defined as a 10 Å sphere from OH atom of Tyr188, and ChemPLP fitness function was used to evaluate the proposed ligand positions.

### 4.4. Biological Evaluation

#### 4.4.1. In Vitro Tests of AChE and BuChE Biological Activity

Inhibitory capacity of compounds on AChE biological activity was evaluated using the spectrometric method of Ellman [18]. Acetyl- or butyrylthiocholine iodide and 5,5-dithiobis-(2-nitrobenzoic) acid (DTNB) were purchased from Sigma Aldrich. Lyophilized BuChE from equine serum (Sigma Aldrich) was dissolved in 0.2 M phosphate buffer pH 7.4 such as to have enzyme solutions stock with 2.5 units/mL enzyme activity. AChE from human erythrocytes (buffered aqueous solution, ≥500 units/mg protein (BCA), Sigma Aldrich) was diluted in 20 mM HEPES buffer pH 8, 0.1% Triton X-100 such as to have enzyme solution with 0.25 unit/mL enzyme activity. In the procedure, 100 μL of 0.3 mM DTNB dissolved in phosphate buffer pH 7.4 was added to the 96 wells plate followed by 50 μL of test compound solution and 50 μL of enzyme (0.05 U final). After 5 min of preincubation at 25 °C, the reaction was then initiated by the injection of 50 μL of 10 mM acetyl- or butyrylthiocholine iodide solution. The hydrolysis of acetyl- or butyrylthiocholine was monitored by the formation of yellow 5-thio-2-nitrobenzoate anion as the result of the reaction of DTNB with thiocholine, released by the enzymatic hydrolysis of acetyl- or butyrylthiocholine, at a wavelength of 412 nm using a 96-well microplate plate reader (TECAN Infinite M200, Lyon, France). Test compounds were dissolved in analytical grade DMSO. Donepezil was used as a reference standard. The rate of absorbance increase at 412 nm was followed every minute for 10 min. Assays were performed in simplicate during three independent tests, with a blank containing all components except acetyl- or butyrylthiocholine, in order to account for non-enzymatic reaction. The reaction slopes were compared, and the percent inhibition due to the presence of test compounds was calculated by the following expression: 100 − (*v_i_*/*v*_0_ × 100) where *v_i_* is the rate calculated in the presence of inhibitor and *v*_0_ is the enzyme activity.

First screening of AChE and BuChE activity was carried out at a 10^−6^ or 10^−5^ M concentration of compounds under study. For the compounds with significant inhibition (≥50%), IC_50_ values were determined graphically by plotting the % inhibition versus the logarithm of six inhibitor concentrations in the assay solution using the GraphPad Prism 6 software.

#### 4.4.2. In Vitro Tests of MAO-A and MAO-B Inhibitory Activity

Inhibition of human recombinant monoamine oxidases A (250 U/mg) and B (59 U/mg; microsomes from baculovirus infected insect cells; Sigma Aldrich) was determined as already described [17], measuring the fluorescence of 4-hydroxyquinoline (exc. 310 nm; em. 400 nm) produced by MAOs in the oxidative deamination of substrate kynuramine. Briefly, compounds were tested at 10 µM (or in the range 3 × 10^−5^–10^−10^ M for determination of IC_50_) in co-incubation with MAO and kynuramine (50 µM) in phosphate buffer 390 mOsm pH 7.4, at 37 °C for 30 min. Assays were performed in triplicate during three independent tests, in 96-well black polystyrene plates (Greiner Bio-One, Kremsmünster, Austria) using an Infinite M1000 Pro plate reader (Tecan, Cernusco s.N., Italy). Inhibition data were obtained as mean ± SD (for single concentration points) or mean ± SEM from 3 independent experiments (for IC_50_), using GraphPad Prism (version 5.00 for Windows; GraphPad Software, San Diego, CA, USA).

## 5. Conclusions

In conclusion, we succeeded, within the frame of this work, in the synthesis of the *trans* diastereoisomers of PADPZ, a structural compromise between DPZ and rasagiline. PADPZ appears endowed with both the activities displayed by these reference compounds and behaves, apparently, in an indifferently manner, as selective AChE and MAO-B inhibitors.

Even if the required level of activities for MTDL are often weaker than those of mono-targeted ligands, due to the synergistic effect MTDL are supposed to carry on, improving the activity of *trans*-PADPZ appears possible through the stereoselective synthesis or separation of its enantiomers. After the proof of interest lying in this original compound this preliminary work brought according to us, this second step, as well as the in vivo evaluation of these enantiomers in animal models of AD, will be quickly undertaken.

## Data Availability

Not applicable.

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
