# Peer review of "First Synthesis of Racemic Trans Propargylamino-Donepezil, a Pleiotrope Agent Able to Both Inhibit AChE and MAO-B, with Potential Interest against Alzheimer’s Disease"

_molecules, 2020, doi:10.3390/molecules26010080_

Round 1

Reviewer 1 Report

The article is very interesting and addresses a relevant topic in public health, I suggest the publication of this work in molecules, it would be good to include more detail of the biological tests, number of replicas, independent tests, wavelength in the measurement of MAO, address theoretically the toxicity of the compounds through some in silico analysis of ADME properties for example

Author Response

it would be good to include more detail of the biological tests, number of replicas, independent tests, wavelength in the measurement of MAO

Cholinesterase tests have been realized in simplicate during three independent tests. This has been indicated in  the experimental part (4.4.1.).

MAO tests have been realized in triplicate during three independent tests.This has been indicated in  the experimental part (4.4.2.).

Wavelenght has been indicated in the experimental part (4.4.2.).

address theoretically the toxicity of the compounds through some in silico analysis of ADME properties for example.

In silico ADME parameters for PADPZ have been calculated and indicated in a new Table 1 (2.2.2.)

All the changes have been highlighted in yellow.

Reviewer 2 Report

The paper "First synthesis of racemic trans propargylamino- donepezil, a pleiotrope agent able to both inhibit AChE and MAO-B, with potential interest against Alzheimer’s disease" by Guieu et al. deals the design, synthesis and functional characterisation of a class of novel Multi-Target Directed Ligands as a potential treatment of Alzheimer’s disease.

The paper has clear focus which is targeted with proper approach, using in-silico docking, synthetic strategy and in-vitro inhibitory activity.

Minor: The authors determined IC50 values for MAO which apparently do not fit with the assumption of a covalent interaction. This result could be explained in part by the dependence of IC50 from the substrate concentration. At high [S]/Km ratio the IC50 will be apparently higher than expected. The authors should report on table 2 the substrates concentrations used in the enzymatic assay.

Author Response

The authors should report on table 2 the substrates concentrations used in the enzymatic assay.

Substrate kynuramine concentration (50 μM) has been added in the text (line 210) and in the experimental part (4.4.2.).

All the changes have been highlighted in yellow.